# The Effects of Endoplasmic Reticulum Stress via Intratracheal Instillation of Water-Soluble Acrylic Acid Polymer on the Lungs of Rats

**DOI:** 10.3390/ijms25073573

**Published:** 2024-03-22

**Authors:** Toshiki Morimoto, Hiroto Izumi, Taisuke Tomonaga, Chinatsu Nishida, Naoki Kawai, Yasuyuki Higashi, Ke-Yong Wang, Ryohei Ono, Kazuki Sumiya, Kazuo Sakurai, Akihiro Moriyama, Jun-ichi Takeshita, Kei Yamasaki, Kazuhiro Yatera, Yasuo Morimoto

**Affiliations:** 1Department of Respiratory Medicine, University of Occupational and Environmental Health, 1-1 Iseigaoka, Yahata-nishi-ku, Kitakyushu 807-8555, Japan; t-morimoto@med.uoeh-u.ac.jp (T.M.); yamasaki@med.uoeh-u.ac.jp (K.Y.); yatera@med.uoeh-u.ac.jp (K.Y.); 2Department of Occupational Pneumology, Institute of Industrial Ecological Sciences, University of Occupational and Environmental Health, 1-1 Iseigaoka, Yahata-nishi-ku, Kitakyushu 807-8555, Japan; h-izumi@med.uoeh-u.ac.jp (H.I.); t-tomonaga@med.uoeh-u.ac.jp (T.T.); n-kawai@med.uoeh-u.ac.jp (N.K.); 3Department of Environmental Health Engineering, Institute of Industrial Ecological Sciences, University of Occupational and Environmental Health, 1-1 Iseigaoka, Yahata-nishi-ku, Kitakyushu 807-8555, Japan; c-nishi@med.uoeh-u.ac.jp; 4Shared-Use Research Center, School of Medicine, University of Occupational and Environmental Health, 1-1 Iseigaoka, Yahata-nishi-ku, Kitakyushu 807-8555, Japan; kywang@med.uoeh-u.ac.jp; 5Department of Chemistry and Biochemistry, The University of Kitakyushu, 1-1, Hibikino, Wakamatsu-ku, Kitakyushu 808-0135, Japan; c1mab005@eng.kitakyu-u.ac.jp (R.O.); c1dab001@eng.kitakyu-u.ac.jp (K.S.); sakurai@kitakyu-u.ac.jp (K.S.); 6Research Institute of Science for Safety and Sustainability, National Institute of Advanced Industrial Science and Technology (AIST), 16-1 Onogawa, Tsukuba, Ibaraki 305-8569, Japan; moriyama-akihiro@aist.go.jp (A.M.); jun-takeshita@aist.go.jp (J.-i.T.)

**Keywords:** polyacrylic acid (PAA), organic compounds, molecular weight, pulmonary toxicity, persistent inflammation, fibrosis, endoplasmic reticulum stress

## Abstract

Polyacrylic acid (PAA), an organic chemical, has been used as an intermediate in the manufacture of pharmaceuticals and cosmetics. It has been suggested recently that PAA has a high pulmonary inflammatory and fibrotic potential. Although endoplasmic reticulum stress is induced by various external and intracellular stimuli, there have been no reports examining the relationship between PAA-induced lung injury and endoplasmic reticulum stress. F344 rats were intratracheally instilled with dispersed PAA (molecular weight: 269,000) at low (0.5 mg/mL) and high (2.5 mg/mL) doses, and they were sacrificed at 3 days, 1 week, 1 month, 3 months and 6 months after exposure. PAA caused extensive inflammation and fibrotic changes in the lungs’ histopathology over a month following instillation. Compared to the control group, the mRNA levels of endoplasmic reticulum stress markers Bip and Chop in BALF were significantly increased in the exposure group. In fluorescent immunostaining, both Bip and Chop exhibited co-localization with macrophages. Intratracheal instillation of PAA induced neutrophil inflammation and fibrosis in the rat lung, suggesting that PAA with molecular weight 269,000 may lead to pulmonary disorder. Furthermore, the presence of endoplasmic reticulum stress in macrophages was suggested to be involved in PAA-induced lung injury.

## 1. Introduction

Chemicals are classified into inorganic and organic substances, and inorganic chemicals, such as asbestos and crystalline silica, are known to cause irreversible interstitial pulmonary fibrotic lesions, such as pneumoconiosis. In contrast, it is thought that organic compounds cause allergic diseases, such as bronchial asthma and hypersensitivity pneumonitis, but they do not directly induce interstitial pulmonary fibrosis. There have been reports in recent years of lung disorders caused by inhalation of organic chemicals, and there has been concern regarding the onset of pulmonary fibrosis caused by organic chemicals. In South Korea, it has been reported that polyhexamethyleneguanidine phosphate (PHMG-p)—an organic chemical used as a humidifier disinfectant—caused severe acute respiratory failure, such as acute respiratory distress syndrome (ARDS), in 5955 people [1,2]. In the United States, multiple cases of acute respiratory failure have been observed with e-cigarettes [3,4]. In Japan, a high rate of progressive lung damage was observed among workers who handle acrylic-acid-based water-soluble polymer compounds (PAA), although it was a small group of 8 out of 82 workers [5]. Furthermore, intratracheal exposure to cross-linked PAA has been reported to induce pulmonary inflammation in rat lung as severe as or equivalent to crystalline silica and asbestos exposure [6,7]. Chemicals, which typically cause lung damage, such as asbestos and crystalline silica, cause lung fibrosis only after more than 10 years of exposure, whereas cross-linked PAA causes lung fibrosis in 2–3 years. It is surprising that lung damage caused by PAA progresses more rapidly than that induced by asbestos and crystalline silica, which have been known to cause lung damage [6].

PAA is a generic term for organic polymer compounds synthesized using acrylic acid as the monomer, and it is an organic material. The polymer’s viscosity and water absorbency are increased when it is cross-linked by a cross-linking agent, and because of this difference in properties, it is used in various applications, such as paints, shampoos and other daily necessities, as well as in additives for food and pharmaceuticals [8]. PAA is considered to be one of the organic substances, which has the least impact on living organisms, and the development of new, related materials is progressing. However, due to reports of pulmonary fibrosis caused by PAA in Japan, concerns have been raised regarding the health effects on consumers, as well as at the manufacturing sites.

The mechanism of lung disorder caused by organic compounds is unknown, but with inorganic compounds, the inhaled chemicals deposit in the lung, causing persistent inflammation and eventually leading to the formation of chronic and irreversible lesions, such as pulmonary fibrosis and tumors [9,10,11,12]. Asbestos, crystalline silica and nanoparticles with high pulmonary toxicity have been reported to cause persistent inflammation in the lungs, leading to lung fibrosis and tumorigenesis [13,14,15].

Endoplasmic reticulum disorder is considered to be involved in persistent lung inflammation. The endoplasmic reticulum plays a crucial role in maintaining biological homeostasis and is responsible for the proper folding of proteins, but when cells are exposed to various internal and external stimuli, proteins, which fail to fold correctly or become denatured, accumulate, resulting in endoplasmic reticulum stress [16,17,18,19]. Studies have indicated that endoplasmic reticulum stress activates multiple signal transduction pathways, leading to various pathological conditions, including inflammation and fibrosis [20,21,22]. Regarding idiopathic pulmonary fibrosis (IPF) in humans, there is a reported correlation between endoplasmic reticulum stress and lung fibrosis [23,24,25,26], and animal models of lung inflammation and fibrosis have demonstrated the induction of endoplasmic reticulum stress [27]. Consequently, it is hypothesized that endoplasmic reticulum stress plays a role in the initiation of lung inflammation and fibrosis.

In the development of inflammation and fibrosis caused by PAA, inflammation and fibrosis appear to progress more significantly compared to those induced by traditional inorganic chemicals, such as asbestos and crystalline silica, indicating a high likelihood that endoplasmic reticulum stress is implicated in the development of inflammation and fibrosis associated with PAA.

In order to investigate lung disorder caused by PAA with different physicochemical characteristics, we performed intratracheal instillation of PAA in a rat model and analyzed the pulmonary inflammation and fibrosis in the lung. We also evaluated whether or not pulmonary inflammation and fibrosis by PAA affect endoplasmic reticulum stress.

## 2. Results

### 2.1. Characterization of PAA

The fundamental characteristics of PAA are summarized in Table 1. The PAA used in our study had weight average molecular weight (Mw) of 2.69 × 10^5^ g/mol and number average molecular weight (Mn) of 2.20 × 10^5^ g/mol according to gel permeation chromatography (GPC) (a Prominence 501 system coupled with Dawn-Heleos-α (Wyatt Technology Europe GmbH, Dernbach, Germany)) using GF-7MHQ (Showa Denko K.K., Tokyo, Japan) with 0.1 M carbonate-bicarbonate buffer as the eluent, respectively.

### 2.2. Relative Lung Weights

There were no significant differences in body weight among all the groups (Figure 1A). The relative lung weight (lung weight/body weight) exhibited a dose-dependent increase during the observation period (Figure 1B).

### 2.3. Cell Analysis and Lactate Dehydrogenase (LDH) Activity in Bronchoalveolar Lavage Fluid (BALF)

Figure 2 shows the results of inflammatory cell number and LDH activity in BALF. In the 0.2 mg exposure groups, there was a statistically significant increase in total cell numbers, the number of neutrophils, percentage of neutrophils, total protein, released LDH activity in BALF from 3 days to 1 week. In the 1.0 mg exposure groups, there was a statistically significant increase in total cell numbers from 3 days to 6 months post-exposure compared to the control group (Figure 2A). The number of neutrophils (Figure 2B) and percentage of neutrophils (Figure 2C) increased significantly from 3 days to 3 months after exposure. The results for total protein, an indicator of vascular permeability, also showed a statistically significant increase during the observation period in the exposed group compared to the control group (Figure 2D). The results of released LDH activity, an indicator of cell damage, also showed a statistically significant increase from 3 days to 1 month and 6 months post-exposure in the exposed group compared to the control group (Figure 2E).

### 2.4. Concentration of CINC in BALF and Concentration of HO-1 in Lung Tissue

Figure 3 shows the concentrations of CXCL1/CINC-1 (CINC-1) and CXCL3/CINC-2 (CINC-2) in BALF and HO-1 in the lung tissue following the intratracheal instillation of PAA. The concentration of CINC-1 increased significantly from 3 days to 1 week in the 0.2 mg exposure groups, and it increased persistently from 3 days to 1 month in the 1.0 mg exposure groups (Figure 3A). The concentration of CINC-2 increased persistently from 3 days to 1 week after exposure in the 0.2 mg exposure groups, and it increased persistently from 3 days to 1 month in the 1.0 mg exposure groups (Figure 3B). The concentration of HO-1 increased significantly only up to 1 week in the 0.2 mg exposure groups, and it increased persistently from 3 days to 1 month in the 1.0 mg exposure groups (Figure 3C). These values tended to decrease with time in the exposed groups.

### 2.5. Micro-CT Imaging

Diffuse or centrilobular infiltration was revealed in the lungs from 3 days to 1 month after exposure in a dose-dependent manner. An improvement in lung infiltrations was observed at 3 months as compared with those at 3 days and 1 month after intratracheal instillation of PAA (Figure 4).

### 2.6. Histopathological Features in the Lung

Histopathological findings in the lung following intratracheal instillation of PAA are shown in Figure 5. The bronchial wall exhibited inflammatory cell infiltration and edema, predominantly neutrophils. There was an increase in type II alveolar epithelium-like cells, fibrin deposition in the alveolar space and the presence of vitreous membrane formation, all of which showed a dose-dependent pattern. These alterations reached their peak at 3 days and persisted up to 1 month, with a gradual tendency to diminish thereafter. Additionally, fibroblasts and collagen fibrils were observed in a dose-dependent manner, peaking at 1 week and persisting, albeit diminishing, up to 6 months (Figure 5A,B). As indicated by the inflammatory scale, there was a notable dose-dependent elevation observed from day 3 onwards, persisting for up to 1 month post-exposure (Figure 5C). The fibrosis score, assessed by the Ashcroft scale, demonstrated a significant dose-dependent increase from 3 days, persisting up to 6 months post-exposure (Figure 5D).

### 2.7. The Relationship between PAA and Endoplasmic Reticulum Stress

The relationship between PAA and endoplasmic reticulum stress is shown in Figure 6. Compared to the control group, the mRNA levels of endoplasmic reticulum stress markers Bip and Chop were significantly increased in the lung tissue in the exposure group (Figure 6A,B), and in line with this, the Bip protein was also significantly increased (Figure 6C). Bip and Chop mRNA levels were significantly increased in the BALF cells of rats at 3 days after intratracheal administration (Figure 6D,E). In fluorescent immunostaining, both Bip and the macrophages exhibited co-localization, and a similar pattern was observed in Chop as well (Figure 6F,G).

## 3. Discussion

The main findings obtained in the present study are as follows: (1) The physicochemical properties of molecular weight (269,000) and non-cross-linked PAA caused inflammation in a dose-dependent manner and induced subsequent lung fibrosis; (2) It was revealed that the expression of endoplasmic-reticulum-stress-related proteins was increased.

In the present study, intratracheal instillation of PAA caused persistent pulmonary inflammation, leading to pulmonary fibrosis. We previously conducted intratracheal instillation studies using the same design to evaluate the inflammatory and fibrotic potential of inorganic materials, where the results showed the pulmonary toxicity of intratracheal instillation of NiO nanoparticles, which have been reported to cause persistent inflammation in the lung [28]. Intratracheal instillation of crystalline silica and asbestos, which are known to cause lung fibrosis, also showed persistent inflammation for more than a month. On the other hand, in the intratracheal instillation of TiO_2_ and ZnO, which have low lung inflammation and fibrotic potential, the inflammation caused by intratracheal instillation was transient [29]. Other reports have also shown that intratracheal instillation of asbestos resulted in persistent inflammation [30] and fibrosis [31]. The findings of those studies are similar to our previous research [6], indicating that chemical substances possessing pro-inflammatory and fibrogenic abilities elicited sustained inflammation, leading to the development of fibrosis with a similar pathological phenotype.

Our previous research showed that the mechanism of this inflammation and pulmonary fibrosis is inflammatory cell infiltration through neutrophil recruitment due to sustained expression of CINC, a neutrophil chemokine, which leads to alveolar damage and increased vascular permeability [6]. It is believed that this causes the inflammation to persist. In the present study, PAA exposure caused lung inflammation as a pathological finding for approximately 1 month, and in conjunction with the pathological finding, CINC, total protein/LDH in BALF and HO-1 in the lung tissue increased persistently.

Intratracheal instillation of PAA under the same intratracheal instillation experimental conditions as in the present study also showed an increase in LDH, total protein, CINC-1 and CINC-2 concentrations in BALF, and an increase in HO-1 in the lung tissue [6]. The number and percentage of neutrophils in BALF were higher than in other substances with high lung inflammatory and fibrotic potential [28]. The above results indicate that the PAA used in this study has inflammatory and fibrotic potential equal to or greater than asbestos and crystalline silica.

The physicochemical properties of the PAA used in this study comprised a molecular weight of 269,000 and no cross-linking. We reported that the inflammatory and fibrotic potential of PAA in the lungs depended on its molecular weight, with PAA with a molecular weight of 600,000 causing persistent inflammation and fibrosis, and PAA with a molecular weight of 30,000 causing only transient inflammation and no fibrosis [32]. The molecular weight in this study was between those two, and the fact that inflammation and fibrosis occurred in the present study indicates that a molecular weight of 269,000 also has the ability to cause inflammation and fibrosis. In previous intratracheal instillation studies, higher molecular weight PAA resulted in more viscous injected suspensions microscopically, while lower molecular weight PAA had poor viscosity. The suspension showed viscosity in this test as well, suggesting that a mechanism mediated by viscosity was involved in the molecular weight dependence of inflammation and fibrosis.

Endoplasmic reticulum stress is a state in which proteins, which are not folded normally, accumulate in the endoplasmic reticulum lumen as defective proteins. Under such circumstances, the endoplasmic reticulum activates the endoplasmic reticulum stress response unfolded protein response (UPR), which acts to eliminate defective proteins. A representative example of this is Bip, a molecular chaperone known as a master regulator of the UPR, which controls this folding. It is thought that when misfolded proteins accumulate and exceed the ability to protect against endoplasmic reticulum stress, endoplasmic-reticulum-stress-induced apoptosis via Chop and other mechanisms eventually occurs.

Exposure to PAA enhanced the expression of Bip and Chop mRNA in the lung tissue, and Bip and Chop mRNA also increased significantly in BALF, which is mainly composed of inflammatory cells. It has been reported that the expression of Bip and Chop is enhanced in vitro and in vivo in asbestos, which has pulmonary inflammatory and fibrotic potential. When A549 cells were exposed to asbestos, Bip and Chop mRNA levels significantly increased 30 min later [33]. Bip and Chop mRNA were significantly elevated in alveolar macrophages of asbestos patients compared to healthy subjects [34].

A protein analysis of Bip and Chop revealed that only Bip was upregulated in the lung tissue, and immunostaining revealed increased expression of Bip and Chop in BALF macrophages. It has been reported that the expression of Bip and Chop proteins is increased in fibrosis models. In human IPF, Bip and Chop protein expression was increased in the lung tissue [35]. In addition, co-staining with macrophages revealed the merging of Bip [36] and Chop [37] in the lung tissue of mice 21 days after intratracheal injection of bleomycin.

In this study, there was no upregulation in Chop expression in the lung tissue (Appendix A), but Chop expression was upregulated in BALF—a result different from previous lesions with strong fibrosis. Although there was no significant difference in the lung tissue as a whole, it was thought that endoplasmic reticulum stress within the alveoli was strong. In particular, the expression of Chop in the macrophages was increased due to endoplasmic reticulum stress, and it is thought that the expression was close to the level of apoptosis. Alveolar macrophage cell death and lung fibrosis have been reported. LPS treatment promoted the apoptosis of alveolar macrophages in silicosis model mice, resulting in a worsening of lung fibrosis [38]. Bip inhibited alveolar macrophage apoptosis and exacerbated lung fibrosis in BLM lung fibrosis model mice [36]. Since excessive infiltration of alveolar macrophages exacerbates pulmonary fibrosis, it is thought that alveolar macrophages promoted apoptosis in order to protect the lungs.

Intratracheal instillation studies can be useful for evaluating the approximate toxicity of inhalable chemicals. A limitation of this study, however, is that its exposure route is not physiological, despite the instillation of PAA of a respirable size; this is not the same as in inhalation studies. Therefore, inhalation studies should also be performed to clarify whether PAA exposure causes lung inflammation and fibrosis and endoplasmic reticulum stress in macrophages.

## 4. Materials and Methods

### 4.1. Sample Polymer

A sample of PAA was synthesized via polymerization of an acrylic acid monomer without a cross-linking agent, as in our previous report [39]. Reversible addition/cleavage chain transfer polymerization was carried out using 4-cyano-4-[(dodecylsulfanylthiocarbonyl) sulfanyl] pentanoic acid (CDP, Sigma Aldrich, St. Louis, MO, USA), α,α′-azobisisobutyronitrile (AIBN, Kanto Chemical Co., Inc., Tokyo, Japan) and t-butyl acrylate. The dissolved solution was transferred to a Schlenk tube and bubbled with nitrogen for 30 min. The reaction was then carried out in an oil bath at 80 °C for 5 h and purified via reprecipitation and filtration in a mixture of water and methanol (2:8). The polymer was dissolved in THF and dried to render t-butyl polyacrylate. The resulting t-butyl polyacrylate was dissolved in dichloromethane (DCM); trifluoroacetic acid (TFA) was added; and the t-butyl group was deprotected by reaction at room temperature for 48 h to render PAA. Hexane was added to the reaction solution to precipitate the PAA, and the solvent was filtered off. The resulting polymer was then dissolved in methanol, dialyzed, purified and lyophilized in 1,4-dioxane to obtain PAA.

### 4.2. Animals

Male Fischer 344 rats (8 weeks old) were purchased from Charles River Laboratories International, Inc. (Kanagawa, Japan) and kept for acclimatization for 4 weeks. They were raised under the same conditions we described previously [6]. Briefly, they were accommodated in the Laboratory Animal Research Center of the University of Occupational and Environmental Health, Japan, with free access to a commercial diet and water. All procedures and animal handling were performed in accordance with the guidelines described in the Japanese Guide for the Care and Use of Laboratory Animals, as approved by the Animal Care and Use Committee, University of Occupational and Environmental Health, Japan (animal studies ethics clearance proposal number: AE18-021).

### 4.3. Intratracheal Instillation

Rats (12 weeks old) received 0.2 mg (0.8 mg/kg BW) or 1.0 mg (4.0 mg/kg BW) of PAA suspended in 0.4 mL distilled water in single intratracheal instillations. The control group received distilled water, and a control group was established for each intratracheal instillation. In the intratracheal instillation study, the rationale for setting low and high doses (0.2 mg and 1.0 mg, respectively) was that the low dose (0.2 mg/rat) represented the minimum dose at which respirable chemicals with high pulmonary toxicity induced lung disorder (persistent inflammation), while the high dose (1.0 mg/rat) was the minimum dose at which respirable chemicals with low pulmonary toxicity did not induce lung disorder. In a previous study, when rats were intratracheally instilled with NiO nanoparticles with high pulmonary toxicity and TiO_2_ nanoparticles with low pulmonary toxicity, NiO nanoparticles induced persistent pulmonary inflammation from the low dose exposure, while TiO_2_ nanoparticles caused only transient inflammation in the lung at the high dose exposure [28]. We also previously examined the biopersistence of TiO_2_ nanoparticles with low toxicity among nanomaterials in rat lungs in an intratracheal instillation study, and neutrophil inflammation in the lung began to delay at doses exceeding 1.0 mg/rat [40]. In other words, lung injury was not observed before clearance was delayed, but lung injury occurred for the first time due to the delayed clearance [41,42]. Therefore, this maximum dose was set assuming human exposure and to avoid overload in the lung.

### 4.4. Animals Following Intratracheal Instillation

Five rats were assigned to each exposure and control group at 3 days, 1 week, 1 month, 3 months and 6 months after intratracheal instillation. Animals were dissected at each time point under anesthesia with isoflurane (Pfizer Japan, Tokyo, Japan) inhalation. Body and lung weights were measured, and then, at autopsy, blood was removed from the abdominal aorta, and the lungs were perfused with normal saline. Briefly, BALF was collected from the right lungs following the removal of blood from the abdominal aorta and perfusion of the right lungs with normal saline. The third lobes of the right lungs were then stored at −80 °C, and the left lungs were inflated and fixed using 10% formaldehyde under a pressure of 25 cm H_2_O for use in histopathological evaluation.

### 4.5. Preparation of Alveolar Macrophages from BALF

The BALF underwent centrifugation for 20 min at 1500 rpm, and the resulting supernatant was discarded, as previously reported [43]. The pellet was then reconstituted in Dulbecco’s Modified Eagle’s Medium (DMEM; Thermo Fisher Scientific, Waltham, MA, USA) supplemented with 10% fetal bovine serum, along with 100 U/mL penicillin and 100 mg/mL streptomycin. Subsequently, cells were cultured on 10 cm plates for 1 h and washed thrice with phosphate-buffered saline.

### 4.6. Cytospin Analysis of Inflammatory Cells and Measurement of LDH in BALF

The BALF pellet and supernatant obtained after centrifugation (400× *g*, 4 °C, 15 min) were used for LDH measurement and cytospin analysis. BALF pellets were processed using the same procedure described in a previous report, and total cell counts were measured on an ADAM-MC (AR BROWN Co., Ltd., Tokyo, Japan). Cells were spread on glass slides using Cytospin (Cyto-Tek^®^ Centrifuge, Sakura Finetek Japan K.K., Tokyo, Japan) and stained with Diff-Quik (Sysmex CO, Kobe, Hyogo, Japan). The numbers of neutrophils and alveolar macrophages were examined under microscopic observation (BX50, OLYMPUS, Tokyo, Japan). The released LDH activity in the BALF supernatant was measured according to the instructions for use of the Cytotoxicity Detection Kit^PLUS^ (LDH) (Roche Diagnostics GmbH, Mannheim, Germany) and estimated using a standard curve obtained from known concentrations of recombinant LDH from rabbit muscle (Roche Diagnostics GmbH, Mannheim, Germany). Regarding sensitivity, it depends on the individual cell type used, with 0.2 × 10^4^ to 2 × 10^4^ cells/well being sufficient for most experiments.

### 4.7. Measurement of Chemokines in BALF and Heme Oxygenase (HO)-1 in Lung Tissue

The levels of concentrations of cytokine-induced neutrophil chemoattractant (CINC)-1 and CINC-2 in BALF supernatant of BALF were determined using ELISA kits, #RCN100 and #RCN200 (R&D Systems, Minneapolis, MN, USA), respectively. All assays were conducted in accordance with the manufacturer’s instructions. The third lobe of the right lung was homogenized, and the protein concentration of the lung homogenate supernatant was measured, as previously reported [44]. The HO-1 concentration in the lung homogenate supernatant was determined using ELISA kit ADI-EKS-810A (Enzo Life Sciences (Farmingdale, NY, USA)).

### 4.8. SDS-PAGE and Western Blotting

The third lobes of the right lungs were homogenized with a T-PER tissue protein extraction reagent (Thermo Scientific Inc., Rockford, IL, USA), including protein inhibitor cocktails (P8340, Sigma-Aldrich, St. Louis, MO, USA) and complete Mini (Roche Diagnostics GmbH, Mannheim, Germany), and then centrifuged (20,400× *g* at 4 °C for 10 min). All specimens underwent sodium dodecyl sulfate-polyacrylamide gel electrophoresis (SDS-PAGE) and were subsequently transferred onto polyvinylidene difluoride (PVDF) membranes. The membrane was initially probed with anti-Chop antibody (1:1000 dilution, #2895, Cell Signaling Technology, Inc., Danvers, MA, USA) and anti-Bip antibody (1:1000 dilution, #3183, Cell Signaling Technology, Inc., Danvers, MA, USA). The detection was carried out using secondary antibodies conjugated with horseradish peroxidase. The signal intensity was quantified using LAS 4000 Mini and Multi Gauge software version 3.0 (Fujifilm, Tokyo, Japan), as detailed in a previous report [45].

### 4.9. Three-Dimensional Micro-CT Imaging

For three out of the five animals in each experimental group, a 3D micro-CT scan was conducted hours to days prior to dissection at each designated observation time. The X-ray 3D micro-CT system used for this purpose was the CosmoScan GX, manufactured by Rigaku Co., Tokyo, Japan. The system was operated under specific conditions, including a scanning time of 4.0 min, an average whole-body exposure of 161.9 mGy per scan, a tube voltage of 90 kV, a tube current of 88 µA and a chest CT with a field of view (FOV) measuring 60 mm × 40 mm (voxel matrix: 512 µm × 512 µm × 512 µm; voxel size: 120 µm × 120 µm × 120 µm). The rats were positioned in the prone posture during the scanning process, and sevoflurane (supplied by Pfizer Japan, Tokyo, Japan) and oxygen were administered through a nose cone. The acquired images were retrospectively gated at both respiratory phases, encompassing both inspiration and expiration.

### 4.10. Quantitative Real-Time Polymerase Chain Reaction

The third lobes of the right lungs (*n* = 5 per group per time point) were homogenized using a QIAzol lysis reagent with a TissueRupotor (Qiagen, Hilden, Germany), as in the previous report [44]. Total RNA from the homogenates was extracted using a miRNeasy Mini Kit (Qiagen, Hilden, Germany), following the manufacturer’s instructions. RNA was quantified using a NanoDrop 2000 spectrophotometer (Thermo Fisher Scientific Inc., Waltham, MA, USA), and the quality of the samples was analyzed by a Bioanalyzer 2100 (Agilent Technologies, Santa Clara, CA, USA). Quantitative real-time polymerase chain reaction (qRT-PCR) was performed, as described previously [44]. Briefly, the total RNA extracted from the lungs at each observation point in each group was transcribed into cDNA (High-Capacity cDNATM Kit, Life Technologies, Tokyo, Japan). qRT-PCR assays were performed using TaqMan (TaqMan Gene Expression Assays, Thermo Fisher Scientific Inc., Waltham, MA, USA), according to the manufacturer’s protocol. Gene expression data were analyzed with the comparative cycle time (ΔΔCT) method using the 7500 Fast Real-Time PCR System. The Assays-on-Demand TaqMan probes and primer pairs included Bip (Assay ID Rn00565250_m1) and Chop (Assay ID Rn00492098_g1). All experiments were performed in a StepOnePlusTM Real-Time PCR Systems (Life Technologies, Tokyo, Japan). All expression data were normalized to endogenous control β-actin expression (Assay ID Rn00667869_m1).

### 4.11. Histopathology

The lung tissue, fixed with formaldehyde, was embedded in paraffin and sectioned to a thickness of 4μm. Subsequently, staining was performed using hematoxylin and eosin (HE), as well as Masson’s trichrome (MT). The evaluation of lung inflammation and fibrosis involved the application of the inflammatory cell infiltration score [44] and the Ashcroft score [46], respectively, in accordance with previous reports [6,44]. To elaborate, the inflammatory cell infiltration score was determined by assessing the extent of inflammatory cell infiltration in the lung tissue, with gradations ranging from none (0) to minimal (0.5), mild (1), moderate (2) and severe (3). The score was calculated using the following formula: Σ (grade × number of animals with that grade). Meanwhile, the Ashcroft score was employed to evaluate lung fibrosis on a scale from 0 (indicating normal lung) to 8 (representing the most severe fibrosis). These grades were cumulatively summed and then divided by the number of fields examined. The histological changes on the slides were analyzed by a pathologist certified by the board.

Immunohistochemical staining was performed, as described previously [47]. Paraffin sections were routinely deparaffinized in xylene and rehydrated in ethanol. The sections were soaked in antigen retrieval solution (Dako, Santa Clara, CA, USA) with an autoclave at 121 °C for 15 min. After treatment with Protein Block Serum-Free (Dako), the sections were incubated with the primary antibodies. The primary antibodies were anti-Chop antibody (#2895, Cell Signaling Technology, Inc., Danvers, MA, USA), anti-Bip antibody (11587-1-AP, Proteintech, Inc.) and MAC2 (CL8942AP, CEDARLANE, Inc., Burlington, ON, Canada). Anti-Chop antibody and anti-Bip antibody were incubated at 4 °C overnight, and MAC2 was incubated for 2 h. The sections were then incubated with the secondary antibodies for 1 h. The sections were then treated with diaminobenzidine and counterstained with hematoxylin. The sections were examined and recorded using a light microscope (VS120, Olympus Corporation, Tokyo, Japan) connected to a digital camera.

### 4.12. Statistical Analysis

Statistical analysis was carried out using IBM^®^ SPSS^®^ software version 28 (IBM Corporation, Chicago, IL, USA). *p* values < 0.05 were considered statistically significant. Unpaired *t*-test was used for two-group comparisons between the 1.0 mg exposure group exposed to PAA samples and the control. Dunnett’s test and Tukey’s honest significant difference test were used to compare the mean values between the three groups: control, 0.2 mg exposure, 1.0 mg exposure (Appendix B).

## 5. Conclusions

Intratracheal instillation of PAA causes lung damage in a molecular-weight-dependent manner, and even PAA with a molecular weight of 269,000 causes lung damage. Furthermore, the presence of endoplasmic reticulum stress in macrophages in PAA-induced lung injury was suggested. For future research, it will be necessary to conduct inhalation studies to clarify whether PAA exposure induces inflammation, fibrosis and endoplasmic reticulum stress in macrophages.

## Figures and Tables

**Figure 1 ijms-25-03573-f001:**
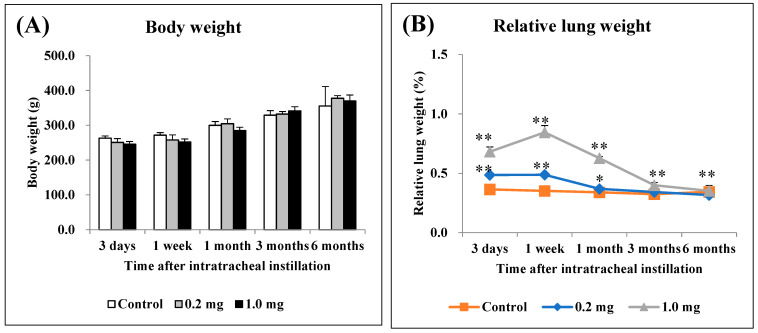
Body weight and relative lung weight after instillation. (**A**) Time course of changes in the body weights of rats in each group. (**B**) Relative weight of the whole lung was calculated as a ratio of whole lung weight (g) to body weight (g) for each rat. The relative lung weight exhibited a dose-dependent increase from 3 days to 3 months. All data are presented as mean ± SE (* *p* < 0.05, ** *p* < 0.01) (Table A1).

**Figure 2 ijms-25-03573-f002:**
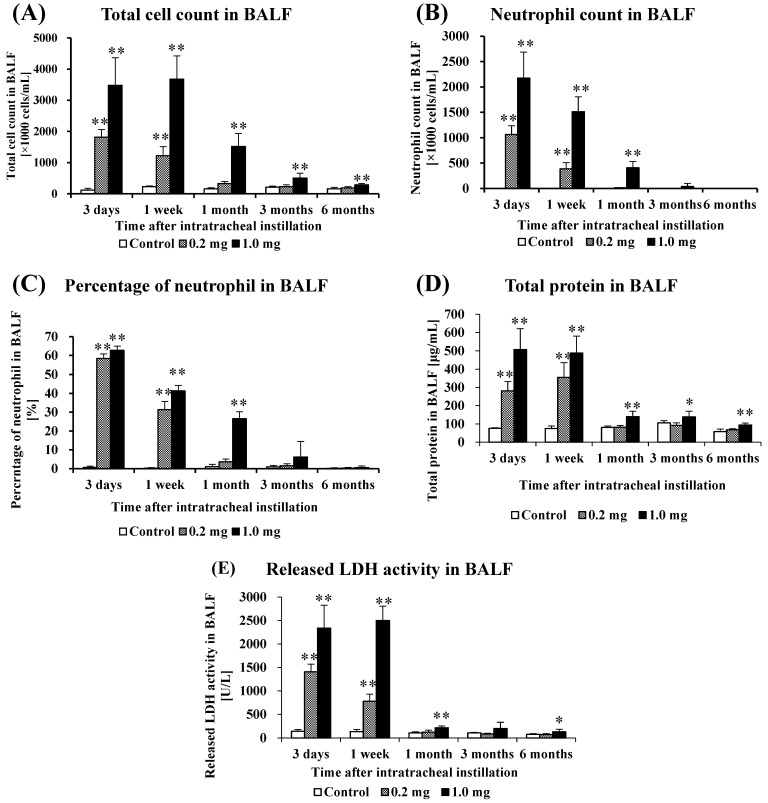
Analysis of cell number, total protein and released LDH activity in BALF following intratracheal instillation. (**A**) Total cell count in BALF, (**B**) Neutrophil count in BALF, (**C**) Percentage of neutrophil in BALF, (**D**) Total protein in BALF, (**E**) Released LDH activity in BALF. All data are presented as mean ± SE (* *p* < 0.05, ** *p* < 0.01) (Table A2).

**Figure 3 ijms-25-03573-f003:**
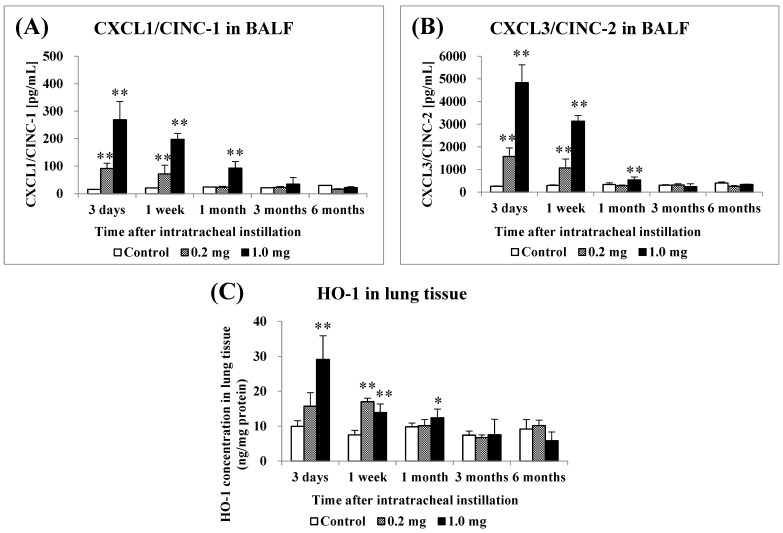
Analysis of chemokines in bronchoalveolar lavage fluid (BALF) and HO-1 in lung tissue following intratracheal instillation. (**A**) CXCL1/CINC-1 in BALF, (**B**) CXCL3/CINC-2 in BALF, (**C**) HO-1 in lung tissue. All data are presented as mean ± SE (* *p* < 0.05, ** *p* < 0.01) (Table A3).

**Figure 4 ijms-25-03573-f004:**
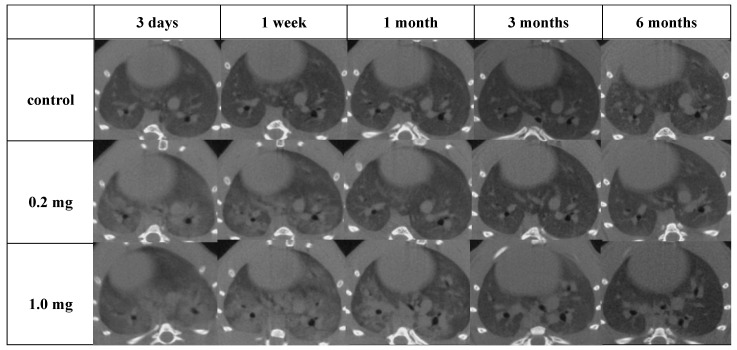
Three-dimensional micro-CT imaging following intratracheal instillation and the manner of viewing the CT image and the relationship between the CT image and the anatomical position. Diffuse or centrilobular infiltration in both lungs was present in a dose-dependent manner at 3 days and 1 month after exposure. The finding persisted for 1 month and improved at 3 months after exposure.

**Figure 5 ijms-25-03573-f005:**
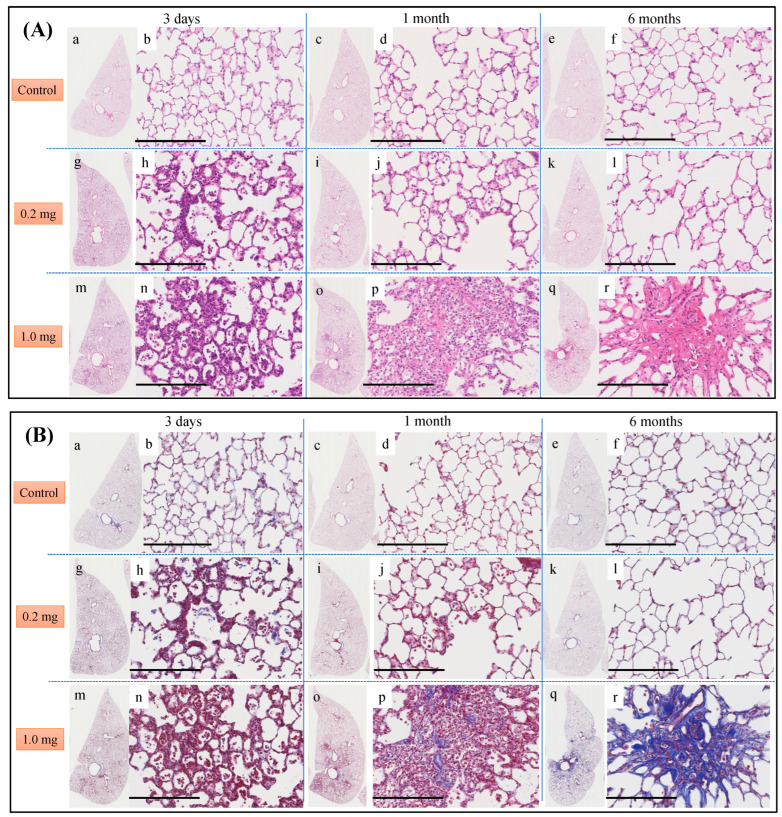
Histological findings after intratracheal instillation (hematoxylin and eosin (HE) and Masson’s trichrome (MT) staining). (**A**) Histological findings with HE staining at 3 days, 1 month and 6 months after intratracheal injection. (**B**) Histological findings with MT staining at 3 days, 1 month and 6 months after intratracheal injection. (**a**–**f**) One of the specimens from the group given distilled water as a control. (**g**–**l**) One of the specimens from the group receiving 0.2 mg of polyacrylic acid (PAA) as the low dose. (**m**–**r**) One of the specimens from the group receiving 1.0 mg of PAA as the high dose. The scale of the black bar is 250 μm. (**C**) Inflammatory scale at each time. (**D**) Ashcroft scale at each time is presented as mean ± SE (** *p* < 0.01, and ‡‡ *p* < 0.01, †† *p* < 0.01) (Table A4).

**Figure 6 ijms-25-03573-f006:**
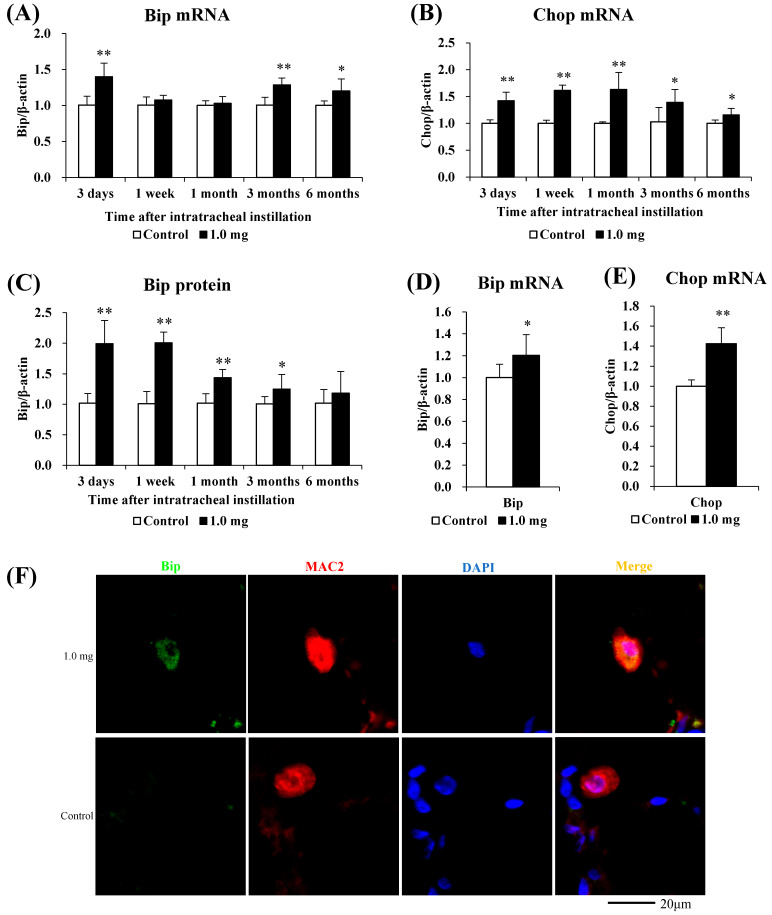
Evaluation of the relationship between PAA and ER stress. (**A**) Bip mRNA, (**B**) Chop mRNA in lung tissue. (**C**) Bip protein in lung tissue using Western blotting. (**D**) Bip mRNA, (**E**) Chop mRNA in cells in BALF. All data are presented as mean ± SE (* *p* < 0.05, ** *p* < 0.01) (Table A5). Unpaired *t*-test was used to appropriately detect differences among the exposed groups. (**F**) Co-localization of Bip and macrophages in fluorescent immunostaining. (**G**) Co-localization of Chop and macrophages in fluorescent immunostaining.

**Table 1 ijms-25-03573-t001:** Physicochemical characterization of the polymers used in the present study.

Physicochemical Characterization	
Structural formula	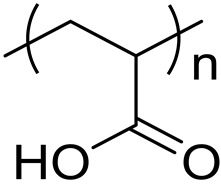
Weight average molecular weight (Mw)	2.69 × 10^5^ g/mol
Number average molecular weight (Mn)	2.20 × 10^5^ g/mol
Poly dispersity index (PDI)	1.22
Degree of cross-linking	None
Radius of gyration (Rg)	49.8 nm

## Data Availability

Data is contained within the article and Appendix A.

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
