# Peer review of "The Effects of Endoplasmic Reticulum Stress via Intratracheal Instillation of Water-Soluble Acrylic Acid Polymer on the Lungs of Rats"

_ijms, 2024, doi:10.3390/ijms25073573_

Round 1

Reviewer 1 Report

Comments and Suggestions for Authors

This manuscript deals with an important topic: the underlying mechanism of the effects of acrylic acid polymers on the lungs. Yet, the manuscript needs wide revisions to be suitable for publication as follows:

Major concerns:

1. The authors have not justified the selection of tested concentrations. The authors mentioned, "This maximum dose was set assuming human exposure and to avoid overload in the lung.” Detailed justification of dose selection and its relation to the real exposure scenarios with references is highly needed. Also, the justification for the duration of the experiment should be explained. Moreover, justify the sample size of 5 rats per group.

2. In the current experiment, the authors used β-actin as a housekeeping gene in gene expression studies. β-actin has potential disadvantages that can compromise its suitability as a stable reference gene. For example, β-actin mRNA levels can be affected by experimental manipulations, including drug treatments, environmental changes, or specific cell culture conditions.  Besides, the β-actin gene undergoes alternative splicing, resulting in the production of different isoforms. These isoforms may have varying expression patterns and functions, which can introduce additional complexity when using beta-actin as a housekeeping gene. To overcome these limitations, validating the stability and expression of potential reference genes under specific experimental conditions is recommended. It is also advisable to consider using multiple reference genes or employing more robust normalization methods, such as normalization against the geometric mean of multiple stable reference genes or using validated reference gene panels.

3. Statistical analysis: The authors mentioned that Dunnett’s tests, Tukey's honestly significant difference test, and unpaired t-test were used appropriately to detect individual differences between those exposed to the cross-linked polyacrylate samples and the controls. The authors should use a model that estimates the main effects of exposure group and time point and potential interactions between these factors.

Other comment:

1. The results should be revised as follows:

- The authors should describe how much change was induced by each concentration of PAA in the results compared with the control. 

- The exact P-value must be provided in the result section to understand statistical significance clearly.

2. Complete information on kits used should be added, including detection range, sensitivity, and inter- and intra-assay.

3. Does data meet the assumption of homogeneity of variances and normal distribution? Clarify if the authors run a homogeneity or normality test.

4. Conclusions: it is better to delete 441-442. Also, the recommendations and further needed studies should be added.

5. The use of abbreviations needs to be revised throughout the manuscript. E.g. line 327, Bronchoalveolar lavage fluid (BALF), then the full term repeated in line 339. Such errors have been repeated throughout the manuscript.

Comments on the Quality of English Language

Good language.

Reviewer 2 Report

Comments and Suggestions for Authors

General comments:

The authors report that intratracheal instillation of PAA induced neutrophil inflammation and fibrosis in the rat lung, indicating that PAA of molecular weight 269,000 may lead to pulmonary disorder. They also demonstrated that the presence of endoplasmic reticulum stress in macrophages was involved in PAA-induced lung injury. All the molecular evidence (mRNA and protein) was used the materials of animal model. Therefore, the animal response is reliable for PAA in regulating lung injury.

Minor comments:

1. The scale of black bar is not mentioned. Please add it in the figure legend.

2. Fig. 6F, 6G: Some images are connected and hard to check. Please make all images separate.

Round 2

Reviewer 1 Report

Comments and Suggestions for Authors

The authors properly addressed all comments.

Author Response

Thank you for reviewing the manuscript.
Your help made the manuscript even better.